# Healthy Behavior and Sports Drinks: A Systematic Review

**DOI:** 10.3390/nu15132915

**Published:** 2023-06-27

**Authors:** Nicolás Muñoz-Urtubia, Alejandro Vega-Muñoz, Carla Estrada-Muñoz, Guido Salazar-Sepúlveda, Nicolás Contreras-Barraza, Dante Castillo

**Affiliations:** 1International Graduate School, University of Extremadura, 10003 Cáceres, Spain; 2Instituto de Investigación y Postgrado, Facultad de Ciencias de la Salud, Universidad Central de Chile, Santiago 8330507, Chile; 3Public Policy Observatory, Universidad Autónoma de Chile, Santiago 7500912, Chile; 4Departamento de Ergonomía, Facultad de Ciencias Biológicas, Universidad de Concepción, Concepción 4070386, Chile; carlaestrada@udec.cl; 5Departamento de Ingeniería Industrial, Facultad de Ingeniería, Universidad Católica de la Santísima Concepción, Concepción 4090541, Chile; 6Facultad de Ingeniería y Negocios, Universidad de Las Américas, Concepción 4090940, Chile; 7Facultad de Economía y Negocios, Universidad Andrés Bello, Viña del Mar 2531015, Chile; nicolas.contreras@unab.cl; 8Centro de Estudios e Investigación Enzo Faletto, Universidad de Santiago de Chile, Santiago 9170022, Chile; dante.castillo@usach.cl

**Keywords:** sports drinks, food supplement, healthy behavior, oral behavior, dental behavior, eating behavior, child, adolescence

## Abstract

This review article aims to systematically identify the relationship between sports drinks and healthy behavior. This systematic literature review was conducted according to the Preferred Reporting Items for Systematic Reviews and Meta-Analyses (PRISMA) guideline criteria, and eligibility criteria were established using the PICOS tool (population, interventions, comparators, outcomes, and study) from about 1000 records of sports drinks articles identified in the various Web of Science Core Collection databases. The literature review stages determined a reduced set of 15 articles relating these drinkable supplements to healthy behavior. This study concludes that water consumption should be emphasized for non-athletes, sports drinks should be labeled to indicate water consumption and carry a warning label, and more randomized clinical trials should be considered to ensure conclusive results for health decision making.

## 1. Introduction

Sports drinks are non-carbonated flavored liquids containing added sugars, minerals, and electrolytes to help replenish the body during strenuous exercise [1]. Some research indicates that its consumption has positive but also negative effects. The medical community is constantly concerned about the health consequences of frequent or excessive consumption of sports drinks [2]. Excessive consumption of sports drinks has been associated with overweight and obesity [3], cardiovascular diseases [4], diabetes [5], and tobacco use [6]. As long as excess sugar remains present in sports drinks, it also poses health risks to the adult population, mainly the prevalence of dental caries [7]. Moreover, the acidity of these beverages (pH < 5.5) and their sugar content of about 4.4 percent (fructose, glucose, and sucrose) contribute to oral health problems, thus posing the same risk of tooth decay as regular sweetened beverages such as fruit juices and carbonated beverages [5,8,9,10]. In non-athletes, especially children and adolescents, sports drinks should not be a daily beverage option and should be consumed only occasionally [11]. In addition, sports drinks contain higher levels of sodium that children do not need in large amounts and can be harmful and increase the risk of high blood pressure and cardiovascular disease [12]. For most children and adolescents, water is sufficient to maintain adequate hydration during physical activity [13]. Sports drinks should only be consumed if the child is participating in strenuous physical activity for more than one hour or if specifically recommended by a physician [14]. At the same time, sports and energy drinks continue to be increasingly popular among adolescents despite the decline of other sugar-sweetened beverages [15]. Water is usually sufficient to maintain adequate hydration. Therefore, sports drinks should not be considered healthy food in general [16]. However, in the case of athletes, research has shown positive effects on their athletic performance. Most people do not need to consume sports drinks to replenish fluids and electrolytes lost during moderate physical exercise, but rather, a specific tool for the hydration and nutrition needs of athletes and people engaged in intense and prolonged physical activity [17,18]. These soft drinks, containing carbohydrates, often sweetened with sugar, serve to restore energy and fluids spent during strenuous exercise, so their use is common among athletes [19]. In most cases, dehydration is associated with loss of sports performance, so it is necessary to maintain a constant source of hydration and available carbohydrates during moderate and intense physical activities [20,21]. During exercise, fluid homeostasis is deregulated when fluid availability is limited, or fluid loss is not adequately recovered [22]. In warm places, dehydration levels will be higher, resulting in greater cardiovascular stress due to the thermal increase [20]. On the other hand, in high-intensity intermittent activities, the use of sports drinks allows for maintaining the intensity of the exercise, contributing to maintaining high levels of circulating glucose and avoiding the depletion of muscle glycogen levels [23]. Sports drinks, both in gel and liquid form, improve hydration by stimulating fluid intake, reabsorption, and retention [20,22], aspects that water alone does not achieve correctly in athletes [24,25]. The sports performance benefits of these beverages are now recognized, and they are considered to contain optimal levels of carbohydrates and electrolytes [26]. However, while most athletes use sports drinks for hydration, the use of beverages containing sugars should be discouraged [27], and even for the average young athlete, the consumption of these beverages may be unnecessary, as they may contribute to negative health outcomes [2]. While the consumption of sports drinks has been associated with higher levels of physical activity and sports engagement, and with the consumption of milk, fruits, and vegetables, it has also been associated with negative health behaviors, such as the intake of higher-calorie foods, and with the consumption of more energy-dense foods [28,29]. Therefore, consumption should be limited to athletes and people engaged in strenuous physical activity and should not be a daily or regular beverage for most people. In general, it is recommended to limit the consumption of sports drinks and opt for healthier options, such as water, to maintain adequate hydration during physical activity [30]. Sports drinks with optimal levels of carbohydrates and electrolytes (4% to 6%) allow for maintaining adequate levels of hydration and blood glucose, generating less cardiac stress and less fluid loss, improving performance in endurance sports [21,31,32,33]. In addition, the constant intake of sports drinks during physical activity and exercise may cause intestinal discomfort despite being the most convenient form of intake [34,35]. Finally, the latest trends in the sports beverage market tend to increase the consumption of sports drinks in the coming years, especially since their consumption is no longer exclusive to athletes and exercisers. This increase in demand would also be boosted by using natural ingredients, especially avoiding the side effects of excess sugar [36]. In other words, consumers are looking for healthier options that help them maintain their energy and performance without the negative side effects of excess sugar [37]. In parallel, companies are striving to create customized sports drinks based on each consumer’s health and fitness data [38]. The sports beverage market is expected to continue to grow in popularity and diversity in the coming years as consumers seek healthier and more personalized options to improve their physical performance. The aim of this study is to systematically review healthy behaviors related to sports drink consumption to answer the following questions: Are there specific relationships according to the studied healthy behaviors? Is the sports drink’s role positively associated with the studied healthy behaviors?

## 2. Materials and Methods

In this review, the Preferred Reporting Items for Systematic Reviews and Meta-Analyses (PRISMA) guidelines [39,40,41] were used, and the PICOS (participants, interventions, comparators, outcomes, and study design) strategy was used to establish the eligibility criteria for the articles [42].

According to the current checklist of the PRISMA guidelines [41], the following quality steps for systematic reviews were verified according to the following items: (1) title, (2) structured abstract, (3) rationale, (4) objectives, (5) eligibility criteria, (6) sources of information, (7) search strategy, (8) selection process, (9) data extraction process, (10a) and (10b) data items, (11) study risk of bias assessment, (14) reporting bias assessment, (16a) and (16b) study selection, (17) study characteristics, (18) risk of bias in studies, (23) discussion, (25) support, (26) competing interests, and (27) availability of data, code, and other materials. The following items were excluded from the PRISMA guidelines due to their non-applicability to the objectives of this review: (12) effect measures, (13) methods of synthesis, (15) certainty assessment, (19) results of individual studies, (20) results of syntheses, (21) reporting biases, (22) certainty of evidence, and (24) registration and protocol. In addition, the initial search for articles was performed using bibliometric procedures [43].

### 2.1. Search Strategy

A set of articles was used as a homogeneous citation base, avoiding the impossibility of comparing indexing databases that use different calculation bases to determine journals’ impact factors and quartiles [44,45,46,47,48], relying on the Web of Science (WoS) core collection, selecting articles published in journals indexed this database, from a search vector on Sports Drinks: {TS = (sport* NEAR/0 drink*)}, without restricted temporal parameters, performing the extraction on 6 January 2023. Only documents typified by WoS as articles were included, regardless of whether these documents had additional parallel typification by WoS.

### 2.2. Eligibility Criteria

The selection of articles was specified based on eligibility criteria: the target population (participants), the interventions (methodological techniques), the elements of comparison of these studies, the outcomes of these studies, and the study designs (the criteria of the PICOS strategy as shown in Table 1).

### 2.3. Study Selection and Data Extraction

In the first step, any duplicates were manually removed. Then, the titles and abstracts of articles were checked for relevance by two researchers. They subsequently, independently from each other, reviewed the full texts of potentially eligible articles. Any disagreements were discussed with a third researcher until consensus was reached.

Then, they excluded meeting abstracts, letters, editorial materials, proceedings papers, and reviews. They excluded articles that were not related to the concept of healthy behavior. Also, articles in Spanish, Portuguese, Japanese, German, Korean, French, and Russian were excluded.

### 2.4. Quality Assessment and Risk of Bias

The Mixed Methods Appraisal Tool (MMAT) scale was used for the assessment of the risk of bias among the included studies. The MMAT scale is a valid measure of the methodological quality of the article. Two authors independently conducted the studies, and a third author was recruited in case of any argument.

Mixed Methods Appraisal Tool (MMAT), a checklist used in systematic reviews based on synthesis of qualitative and quantitative evidence, includes criteria for the evaluation of mixed studies; it defines the study category, and 7 items are applied according to a score from zero to one, to obtain a final percentage mean. Studies are considered as high quality > 75%, moderate quality 50–74%, and low quality < 49%. Studies with values below 75% were excluded from the category analysis and discussion [50].

## 3. Results

The search vector extracted a total of 991 records without repetitions from seven different databases of the Web of Science Core Collection (i.e., SSCI; SCI-EXPANDED; ESCI; CPCI-SSH; CPCI-S; BKCI-SSH; A&HCI) from 1984 to 2023. Excluding records according to the PICOS tool (see Table 1), non-article records (174) and non-English-language articles (39) resulted in 778 records for screening (details in Appendix A). In addition, 763 articles not related to the healthy behavior concept are also excluded. Thus, reducing the corpus analyzed to 15 full-text articles in English retrieved and screened using the selection criteria defined in the previous section (See Figure 1).

Using the PRISMA guidelines, fifteen articles were selected [41]. Table 2 shows details, which are authors, publication sources, indexation, citations, and study designs. In addition, Table 3 details each 15 articles’ eligibility criteria using (MMAT) Mixed Methods Appraisal Tool.

Then, the studies included in this review have been classified according to their main topics in three categories, as shown in Table 4.

Thus, the articles analyzed mention three specific types of healthy behaviors related to sports drinks consumption: healthy oral–dental behavior, healthy eating behavior (overweight/obesity), and healthy eating behavior (nutritional functional). The results are detailed below.

### 3.1. Results of Healthy Oral–Dental Behavior and the Use of Sports Drinks

Four categorized studies focus on children [7,57,60,61], establishing as main conclusions that: (a) there are racial–ethnic disparities in its consumption to the detriment of Latin and African American children; (b) in the dental caries occurrence due to its consumption, the child’s health behavior prevails over that of the mother, although good parenting behaviors of mothers could improve the dental health of their children; (c) healthy behaviors are quite stable as long as they are assumed at younger ages; (d) the polarization of the differences in healthy oral health behaviors in adolescents can start from there, even independent of the parents’ occupational level. In cases such as adult men [6], energy/sports drink consumption is positively associated with smoking and ta-smoking.

Also, it is important to note that the only randomized clinical trial among all those selected belongs to this category [60]. This type of randomized design has three advantages over other selection methods for intervention and control groups: (1) randomization eliminates the possibility of bias in the participant assignment for both types of groups; (2) randomization tends to produce comparable groups since prognostic factors (known and unknown) and other participant characteristics will be balanced among the groups to be compared; (3) they ensure the validity of statistical significance tests, independent of the balance in prognostic factors among the randomized groups, avoiding making further assumptions about the comparability of groups and the statistical model adequacy [62] (p. 92).

### 3.2. Healthy Behavioral Outcomes for Overweight and Obesity and the Use of Sports Drinks

The categorized studies focus on adolescents, which conclude the following: (a) the existence of racial–ethnic disparities in their consumption, to the detriment of Latino and African American children, is also related to this healthy behavior category [7]; (b) the decrease in daily consumption, in general, of sports drinks and their persistent popularity among adolescents who at least consume them on a weekly basis, of concern is the daily use by adolescents who watch television more than 2 h per day; (c) weight misperception among overweight/obese adolescents and its effect on weight-related health-enhancing behaviors, which may partially explain the protective effect of weight misperception on weight gain over time; (d) the association between sports drink consumption and increased moderate to vigorous physical activity (MVPA), and adolescent recall of consumption only after prolonged vigorous activity; (e) the need for interventions to reduce sports drink consumption to address a range of unhealthy behaviors; (f) the relationship between lower youth consumption of water, given their access and food safety concerns with increased consumption of sugar-sweetened beverages.

### 3.3. Results of Healthy Nutritional Behavior and the Use of Sports Drinks

These results present, in addition to the two previous categories, only two studies different from those considered above: (a) one regarding closed campus policies and their effects on reducing consumption of sugar-sweetened beverages (SSB; soft drinks and sports drinks; coffee or tea sugar-sweetened beverages), although students who are accustomed to purchasing beverages may switch from off-campus to on-campus alternatives, ensuring healthy school food environments is relevant [53]; (b) another study regarding weight modification (WM) in army soldiers and its relationship with multiple health behaviors, including tobacco use, physical fitness score, self-perceived health, and eating behavior. As a result, nutritional supplementation (NS) was found to add up in this population regardless of the weight modification (WM) goal [58].

## 4. Discussion

The systematic review presented in this manuscript examines 991 scientific articles from seven databases of the Web of Science Core Collection on healthy behaviors associated with the consumption of energy drinks, performing a screening that results in 15 articles being reviewed in depth, which allows an initial categorization into three groups that are healthy oral–dental behavior, healthy eating behavior (overweight/obesity) and healthy eating behavior (nutritional functional), the latter being discarded because it does not allow to see significant differences with the category referred to overweight/obesity.

Regarding the first group, healthy oral–dental behavior, this systematic review coincides with other previous reviews regarding the association between oral–dental health and sports drinks [63,64,65,66,67]. Some of these previous reviews also report a negative effect of sports drinks consumption on oral–dental health, such as that of Nijakowski et al. [63], who state that “regular physical activity was associated with an increased risk of dental erosion, especially under the influence of frequent consumption of sports drinks”. Other results may be even more compelling in indicating that high-sugar beverages (including sports drinks) lack nutritional value and contribute to the burden of dental disease in all age groups. In particular, the situation in children is of concern, given that manufacturers of sports and energy drinks have chosen to target children in their marketing campaigns and promote a misleading association between their products, healthy lifestyles, and the athletic process [65]. The information on food specifications and healthful qualities provided by labels in this regard takes on special value, although when it has its greatest effect on young, informed consumers [68,69,70,71].

Dental erosion caused by the ingestion of sports drinks and other foods has increased and cannot be ignored by clinical dental practice [7,10]. Although the review by Coombes [66] indicates that “the use of sports drinks does not provide benefits over water”, these drinks are useful to prevent dehydration and heat stroke due to the increase in temperature and sweating rate [62,67]. Especially relevant in intense or prolonged activities, where maintaining adequate hydration [20,21], a correct number of electrolytes [17,18,67], feeling cool [67], and an optimal level of circulating carbohydrates to prevent muscle glycogen depletion and maintain performance are crucial [23,26,31,32]. However, water alone is not sufficient to achieve this in athletes [24,25,72,73].

During intense activities, other reviews recommend ingesting fluids with a carbohydrate concentration of 5–10% and up to 10–12% to reduce the total amount of fluids consumed and maintain a correct hydration level [67]. Other studies support 500 mL/h of fluids during exercise [67,74,75] to lose no more than 2–3% of body weight in sweat [76]. However, the choice of fluid will depend on the characteristics and intensity of the activity, according to Burke and Read [67]. Hydration levels may be self-regulating, with a good initial hydration level, low thirst sensation, and pale urine color. Urine concentration, especially in the early morning, can overcome an unreliable thirst sensation or when external factors such as psychological stress or repeated food intake can dull the thirst sensation and is a simple way to assess hydration status [77,78]. Therefore, other dental protection mechanisms should be considered to prevent acid erosion in the athlete population [64].

As a contribution of our work, in the relationship between the consumption of sports drinks and oral–dental health, we highlight the incorporation of the concept of “healthy oral–dental behavior”, given that previous reviews [63,64,65,66,67] do not emphasize this, exploring only aspects related to health.

On the other hand, regarding our results identifying a relationship between healthy eating behavior in overweight and obese conditions and the use of sports drinks, there is an agreement with previous systematic reviews [79,80,81,82]. Some of these reviews [80,81,82], given the prevalence of overweight, obesity, and diabetes, consider beverages to be an important source of caloric intake, collecting research that has documented how caloric beverages increase the risk of obesity. They also identify sports drinks, based on calorie content, nutritional value, and health risks, at level 5 of 6, comprising drinks with high-calorie content and limited health benefits (fruit juices, whole milk, and fruit smoothies with sweetened honey; alcoholic and sports drinks), with level 6 being the least healthy.

In addition, sports drinks have high levels of sodium, which can lead to health problems such as high blood pressure and heart disease [4,12]. Therefore, it is recommended to limit the consumption of sports drinks by non-athletes, especially in children and adolescents [11]. In this sense and aligned with other reviews, we believe that it is necessary to improve the labeling of sports drinks to generate awareness of the deleterious effects of their consumption in the non-athlete population [12,44]. Emphasizing the consumption of water, which in most cases is more than sufficient to hydrate and refresh during exercise and physical activity [13].

Previous reviews [79,80,81,82] have related sports beverage consumption to dietary health (overweight/obesity), but they have not studied the concept of healthy eating behavior, which we highlight in our review. Therefore, we reinforce, as a contribution of this review, the incorporation of the concept of healthy behavior and not only the concept of health.

One limitation of this study is the use of seven databases from the Web of Science Core Collection (WoS), whose data have been added as Appendix A. The exclusive use of databases included in WoS allows homogeneous search criteria (a unique search vector) and the possibility of comparing the quality of the documents analyzed, in terms of citations, given that the bibliographic databases have different criteria for inclusion of documents and, therefore, different counting bases to establish their metrics [44,45,46,47,48], generating variations in the results of systematic literature reviews. As future lines of research, in view of these findings relating to the child (oral–dental behavior) and adolescent (unbalanced eating behavior) populations, which we could classify as unhealthy consumption behaviors, we propose that in the future, we should delve more deeply into other reviews relating to their eating behaviors that could put their health at risk as well as addressing other effects of nutritional eating behavior in the different stages of the life cycle and their effects on health.

Finally, to strengthen healthy eating behaviors and provide relevant information on the potential harms of sports drinks in the non-athlete population [12,83], more randomized control trials are needed. These studies will provide accurate and reliable data [84] on the effects of healthy eating behaviors [85] by randomly assigning participants into treatment and control groups. Conducting more randomized control trials will contribute to the generation of solid evidence to support the adoption of healthy eating and nutritional behaviors and informed decision making for the protection of public health [84,85,86]. Furthermore, Spodick [87] points out that scientifically the randomized controlled trial is the most powerful way of proving that an outcome is attributable exclusively to the trial treatment. On the other hand, randomization implies an equal chance of not obtaining a trial treatment, making it the most ethical design.

## 5. Conclusions

The evidence presented in this systematic review suggests emphasizing water consumption in non-athletes to prevent the harmful effects of sports drink consumption on oral–dental health and to prevent dehydration and heat stroke.

On the other hand, our findings suggest to business decision makers that their marketing campaigns for this type of food should indicate the risks of its consumption, and we propose that these products improve their composition and nutritional contribution. Furthermore, public policymakers should focus more on the consequences of the consumption behavior of sports drinks to prevent and remedy the consequences on oral–dental health and overweight/obesity.

Finally, we believe unique randomized clinical trials would mobilize research leaders and funding decision makers to increase efforts and incentives to expand studies of this type.

## Figures and Tables

**Figure 1 nutrients-15-02915-f001:**
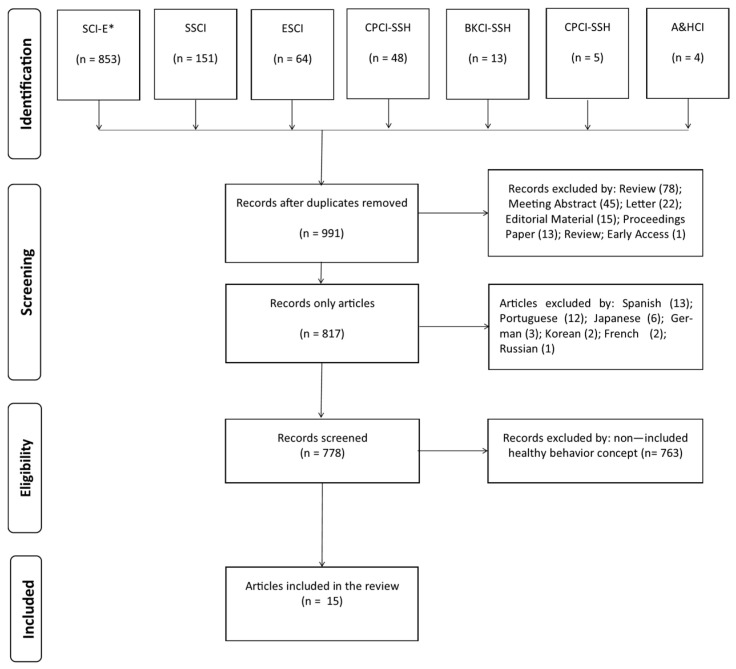
Preferred Reporting Items for Systematic Reviews and Meta-Analyses (PRISMA) analysis flow. SCI-E* = Science Citation Index Expanded; SSCI = Social Sciences Citation Index; ESCI = Emerging Sources Citation Index; CPCI-SSH = Conference Proceedings Citation Index—Social Science and Humanities; CPCI-S = Conference Proceedings Citation Index—Science; BKCI-SSH = Book Citation Index—Social Sciences and Humanities; A&HCI = Arts and Humanities Citation Index.

**Table 1 nutrients-15-02915-t001:** Eligibility criteria using PICOS (participants, interventions, comparators, outcomes, and study design).

PICOS	Description
Population	Persons, without age restriction, who consume or could potentially consume sports drinks
Interventions	Application of questionnaires, interviews, focus groups, or health and motor physical exams in humans
Comparator	Contains the concept behavior and derivations (including in British English, “behaviour”) in title, author keywords, keywords plus ^®^ or abstract of the article.
Outcomes	Sports drink consumption behaviors and effects on health and wellbeing.
Study designs	No a priori restrictions. Quantitative, qualitative, and mixed-study types were included (under MMAT quality criteria) [49].

**Table 2 nutrients-15-02915-t002:** Selected articles by PRISMA guidelines.

Authors	Affiliations	Journal	Publication Year	Sample	WoS Index	Pubmed (Y/N)	Times Cited, WoS Core	Category of Study Designs
Zoellner et al. [51]	Univ. Virginia; Univ. Dayton; Univ. Syst. Ohio; Miami Univ.	Matern. Child Health J.	2022	*n* = 202	SSCI	Y	0	Quantitative descriptive
Anselma et al. [52]	Univ. Amsterdam; Vrije Univ. Amsterdam; Qinghai Norm. Univ.	Health Educ. Behav.	Early Access	t0 = 700, t1 = 538, t2 = 632	SSCI	Y	0	Quantitative non-randomized
Patte et al. [53]	Brock Univ.; Univ. Waterloo	Health Promot. Chronic Dis. Prev. Can.-Res. Policy Pract.	2021	n1 = 60,610, n2 = 134	SCI-E	Y	2	Quantitative non-randomized
Beck et al. [7]	Univ. Calif. Syst.; Univ. Calif. San Francisco; Stanford Univ.	Public Health Nutr.	2020	t1 = 4901, t2 = 3606	SCI-E + SSCI	Y	5	Quantitative non-randomized
Gallagher et al. [27]	Univ. London; Univ. Coll. London	Br. Dent. J.	2019	*n* = 352	SCI-E	Y	12	Quantitative descriptive
Hess et al. [54]	Univ. New Mexico; West. Oregon Univ.	Public Health Nutr.	2019	*n* = 40	SCI-E + SSCI	Y	12	Qualitative
Cordrey et al. [55]	Northwell Health; North Shore Univ. Hosp.; Steven and Alexandra Cohen Child. Med. Cent. N.Y.; Univ. Syst. Ohio; Ohio State Univ.; Nationwide Child. Hosp.; Res. Inst. Nationwide Child. Hosp.; Hofstra Univ.	Pediatrics	2018	t1 = 11,458, t2 = 15,624	SCI-E	Y	11	Quantitative descriptive
Hahn et al. [56]	Univ. Michigan Syst.; Univ. Michigan	BMC Public Health	2018	*n* = 4383	SCI-E + SSCI	Y	15	Quantitative non-randomized
Pakkila et al. [6]	Univ. Oulu; Univ. Gothenburg; Univ. Helsinki; Helsinki Univ. Central Hosp.	Acta Odontol. Scand.	2017	*n* = 3420	SCI-E + SSCI	Y	5	Quantitative non-randomized
Larson et al. [28]	Univ. Minnesota Syst.; Univ. Minnesota Twin Cities; Duke Univ.	J. Nutr. Educ. Behav.	2014	*n* = 2793	SCI-E + SSCI	Y	91	Quantitative descriptive
Tolvanen et al. [57]	Univ. Oulu; Univ. Turku	Acta Odontol. Scand.	2014	*n* = 1691	SCI-E + SSCI	Y	0	Quantitative descriptive
Austin et al. [58]	U.S. Dep. Energy (DOE); Oak Ridge Inst. Sci. and Educ.; U.S. Dep. Def.; U.S. Army	Int. J. Sport Nutr. Exerc. Metab.	2013	*n* = 990	SCI-E	Y	7	Quantitative descriptive
Cunningham et al. [59]	Cent. Dis. Control Prev.—USA	Matern. Child Health J.	2012	*n* = 1522	SSCI	Y	24	Quantitative descriptive
Tolvanen et al. [60]	Univ. Oulu	Community Dentist. Oral Epidemiol.	2010	*n* = 1691	SCI-E + SSCI	Y	12	Quantitative randomized clinical trials
Kawashita et al. [61]	Nagasaki Univ.	J. Public Health Dent.	2009	*n* = 396	SCI-E + SSCI	Y	11	Quantitative descriptive

**Table 3 nutrients-15-02915-t003:** Eligibility criteria using (MMAT) Mixed Methods Appraisal Tool.

Authors	Journal	Publication Year	Category of Study Designs	S1	S2	1,1	1,2	1,3	1,4	1,5	2,1	2,2	2,3	2,4	2,5	3,1	3,2	3,3	3,4	3,5	4,1	4,2	4,3	4,4	4,5	Quality
Zoellner et al. [51]	Matern. Child Health J.	2022	Quantitative descriptive	1.0	1.0	-	-	-	-	-	-	-	-	-	-	-	-	-	-	-	0.8	0.3	0.8	0.8	0.7	77%
Anselma et al. [52]	Health Educ. Behav.	Early Access	Quantitative non-randomized	1.0	1.0	-	-	-	-	-	-	-	-	-	-	0.6	1.0	0.7	0.7	1.0	-	-	-	-	-	86%
Patte et al. [53]	Health Promot. Chronic Dis. Prev. Can.-Res. Policy Pract.	2021	Quantitative non-randomized	1.0	1.0	-	-	-	-	-	-	-	-	-	-	1.0	1.0	1.0	0.8	1.0	-	-	-	-	-	97%
Beck et al. [7]	Public Health Nutr.	2020	Quantitative non-randomized	1.0	1.0	-	-	-	-	-	-	-	-	-	-	1.0	1.0	0.6	0.5	1.0	-	-	-	-	-	87%
Gallagher et al. [27]	Br. Dent. J.	2019	Quantitative descriptive	1.0	1.0	-	-	-	-	-	-	-	-	-	-	-	-	-	-	-	0.4	0.0	0.5	0.3	0.5	53% *
Hess et al. [54]	Public Health Nutr.	2019	Qualitative	1.0	1.0	0.7	0.6	0.6	0.4	0.5	-	-	-	-	-	-	-	-	-	-	-	-	-	-	-	54% *
Cordrey et al. [55]	Pediatrics	2018	Quantitative descriptive	1.0	1.0	-	-	-	-	-	-	-	-	-	-	-	-	-	-	-	0.4	0.0	0.5	0.5	0.4	54% *
Hahn et al. [55]	BMC Public Health	2018	Quantitative non-randomized	1.0	1.0	-	-	-	-	-	-	-	-	-	-	1.0	1.0	0.8	0.8	0.7		-	-	-	-	90%
Pakkila et al. [6]	Acta Odontol. Scand.	2017	Quantitative non-randomized	1.0	1.0	-	-	-	-	-	-	-	-	-	-	1.0	1.0	0.7	0.7	0.7		-	-	-	-	87%
Larson et al. [28]	J. Nutr. Educ. Behav.	2014	Quantitative descriptive	1.0	1.0	-	-	-	-	-	-	-	-	-	-	-	-	-	-	-	0.8	0.2	0.8	0.8	0.7	76%
Tolvanen et al. [57]	Acta Odontol. Scand.	2014	Quantitative descriptive	1.0	1.0	-	-	-	-	-	-	-	-	-	-	-	-	-	-	-	0.8	0.2	0.7	0.8	0.9	77%
Austin et al. [58]	Int. J. Sport Nutr. Exerc. Metab.	2013	Quantitative descriptive	1.0	1.0	-	-	-	-	-	-	-	-	-	-	-	-	-	-	-	0.7	0.4	0.8	0.8	0.7	77%
Cunningham et al. [59]	Matern. Child Health J.	2012	Quantitative descriptive	1.0	1.0	-	-	-	-	-	-	-	-	-	-	-	-	-	-	-	1.0	0.4	0.9	0.9	0.9	87%
Tolvanen et al. [60]	Community Dentist. Oral Epidemiol.	2010	Quantitative randomized clinical trials	1.0	1.0	-	-	-	-	-	1.0	0.6	0.8	0.9	0.9	-	-	-	-	-	-	-	-	-	-	76%
Kawashita et al. [61]	J. Public Health Dent.	2009	Quantitative descriptive	1.0	1.0	-	-	-	-	-	-	-	-	-	-	-	-	-	-	-	0.9	0.2	0.7	0.8	0.8	77%

* Study is considered as moderate quality 50–74%, and was excluded from the category analysis, and discussion.

**Table 4 nutrients-15-02915-t004:** Classified according to their main topics in 3 categories.

Authors	Oral/ Dental/ Caries	Overweight/ Obesity	Nutritional	Total
Zoellner et al. [51]	-	-	-	0
Anselma et al. [52]	-	X	X	2
Patte et al. [53]	-	-	X	1
Beck et al. [7]	X	X	-	2
Hahn et al. [56]	-	X	X	2
Pakkila et al. [6]	X	-	-	1
Larson et al. [28]	-	X	X	2
Tolvanen et al. [57]	X	-	-	1
Austin et al. [58]	-	-	X	1
Cunningham et al. [59]	-	-	X	1
Tolvanen et al. [60]	X	-	-	1
Kawashita et al. [61]	X	-	-	1
Total	5	4	6	

## Data Availability

Data availability in the Appendix A.

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
