# Peer review of "Healthy Behavior and Sports Drinks: A Systematic Review"

_nutrients, 2023, doi:10.3390/nu15132915_

Round 1

Reviewer 1 Report (Previous Reviewer 2)

In terms of structure, the quality of the article has improved considerably from its original form. However, there are still some points that need to be improved.
First of all the pages are not set to the same size. The numbering of the lines is only up to the middle of the article, more precisely up to line 176.
Lines 109-121... you made an enumeration of the type: "1 (title), 2 (structured 110 abstract), 3 (rationale), 4 (objectives), 5 (eligibility criteria).... *. I recommend you put the numbers in parentheses, not the content (for example: (1) title, (2) structured abstract, (3) rationale... ).
Please also discuss the limitations of the study..
The number of references has improved. The article had 87 references.

Author Response

Dear reviewer 1, we have corrected your comments. And highlighted in light blue the changes in response to the Reviewer team.
We appreciate your assessment of the improvement in the quality of the article from its original form. 
1) Pages are not the same size: The page size in the document has been standardized.
2) The line numbering only goes up to the middle of the article. Line numbers have been incorporated throughout the document.
3) Lines 109-121.... I recommend that you put the numbers in parentheses, not the content.... : The numbering for the PRISMA criteria has been adjusted according to your request.
4) We have commented the limitation by adding 5 new references.
5) We appreciate that you indicate that the number of references has improved. With the new references added in this round, we have increased again from 87 to 92 references.

Reviewer 2 Report (New Reviewer)

This systematic review is well-conducted and written.

I advise the authors to correct the following:

Line 39: diabetes [5],

Line 45: [5,8-10]

Line 88: sports [21,31-33].

Line 106: [39-41]

Line 109/121: I think this part is unnecessary. The authors are still free to leave it.

After page 5/17, the numbering resumes from 1/17; check the pagination.

Pag. 8, section 3.1: Where in the text there are lists (a, b, c, d or 1, 2, 3), it is advisable to place a ";" between one point and another.

Pag. 9, section 3.2, 3.3: Where in the text there are lists (a, b, c, d or 1, 2, 3), it is advisable to place a ";" between one point and another.

Pag. 10: [68-71].

Author Response

Dear reviewer 2, we have corrected your first round comments to this article resubmission. And highlighted in light blue the changes in response to the review team.
Line 39: Improved punctuation. "," added.
Line 45: Corrected citation. "-" corrected.
Line 88: Corrected scoring and citation. "-" corrected.
Line 106: Citation corrected. "-" corrected.
Line 109/121:We have added new adjustments to this section per reviewer 1's request.
Check pagination: We have re-adjusted the pagination of the document.
P. 8, section 3.1: It is advisable to place a ";" between periods. We have adjusted the punctuation lines 215 to 230.
P. 9, section 3.2, 3.3: It is advisable to place a ";" between a period and another.
We have adjusted the punctuation lines 236 to 254.
P. 10: Citation corrected. "-" corrected. Line 280.

This manuscript is a resubmission of an earlier submission. The following is a list of the peer review reports and author responses from that submission.

Round 1

Reviewer 1 Report

Dear authors,

You made a very good review, a very complex work Based on adequate methodology, well described. The review must be improved by highlight în more the utility, the relevance, the managerial implications and future perspectives.

Author Response

Dear reviewer, thank you for your overall assessment of the manuscript and the methodology used. 
We have highlighted its usefulness, relevance and implications for decision makers. We have also highlighted and increased future research lines.

Reviewer 2 Report

This is a systematic review about healthy behavior related to the sports drinks consumption.
You described the checklist of the PRISMA guidelines. I don't think it is necessary to describe them, the important thing is to apply them all. All the description and table no. 1 will deviate from your topic, should be erased, only the diagram should remain.
Line 131...please correct *The following types of documents were included only articles and parallel document types."
You have a paragraph with the Author Contributions, it is not necessary to specify in the text what each author did.
Line 148... apply correction in "Then articles excluded in language Spanish, Portuguese, Japanese, German, Korean, French, Russian."
I recommend you to expand the discussions and make a separate paragraph with the conclusions, otherwise the review is too short, should exceed at least 4000 words...
The references are appropriate, the article presents 70 references, being up to date. Some references have the years in bold, others don't.

Author Response

Dear reviewer, very grateful for your comments. We explain the required adjustments below (changes are highlighted in light green in the manuscript):

1) Regarding the PRISMA checklist, Table 1 and diagram (Figure 1). 
The manuscript you have evaluated was adjusted according to precise indications of the Editor, in a previous and formal stage of admissibility evaluation, in which we were asked for minimum requirements for the "Materials and Methods" section.
Thus, what was submitted for your evaluation was adjusted according to 2 examples of systematic reviews previously published in the journal that we were instructed to follow.
If there is a pronouncement in this regard from the Editor, we have no objection to eliminate what you indicate.

2) Line 131...please correct. We have corrected the wording.
3) In reference to the identification of specific author's tasks (sections 2.3 and 2.4), as a complement to the CREDIT protocol used in the section on author's contributions.
Again, our manuscript was adjusted in this way by editorial request, according to the 2 examples of formats indicated below.
If there is a pronouncement in this respect from the Editor, we have no objection to eliminate what you indicate.
4) Line 148... apply correction.  We have corrected the wording.
5) We have separated the conclusion and discussion sections.
6) We have expanded the discussion.
7) "should exceed at least 4000 words...". The document submitted for evaluation was 7890 words, the document submitted now exceeds 8100. The number recorded on the platform (3657 words) is a system error that the publisher could not correct.
8) The references that did not have the years in bold have been corrected.

Round 2

Reviewer 1 Report

Dear authors,

I noticed you made improvements. Please place the conculions after the discussion section.

Author Response

Dear reviewer, very grateful for your comments. We explain the required adjustments below (changes are highlighted in light green in the manuscript):

RESPONSE TO ROUND2:
We have reordered discussion and conclusions. 
The conclusion has been summarized.

Reviewer 2 Report

First, conclusions are the last part of a paper, not discussions. The conclusions must be the shortest and represent the "ideas that the reader stays with". You have mixed discussions with conclusions and nothing is clear anymore. From my point of view, the article cannot be accepted in this form. I also stand by my previous comments.

Author Response

Dear reviewer, very grateful for your comments. We explain the required adjustments below (changes are highlighted in light green in the manuscript):

RESPONSE TO ROUND1:
1) Regarding the PRISMA checklist, Table 1 and diagram (Figure 1). 
Table 1 has been deleted.
2) Line 131...please correct. We have corrected the wording.
3) In reference to the identification of specific author's tasks (sections 2.3 and 2.4), as a complement to the CREDIT protocol used in the section on author's contributions.
We have eliminated author identification.
4) Line 148... apply correction.  We have corrected the wording.
5) We have separated the conclusion and discussion sections.
6) We have expanded the discussion.
7) "should exceed at least 4000 words...". The document submitted for evaluation was 7890 words, the document submitted now exceeds 8100. The number recorded on the platform (3657 words) is a system error that the publisher could not correct.
8) The references that did not have the years in bold have been corrected.

RESPONSE TO ROUND2:
We have reordered discussion and conclusions. 
The conclusion has been summarized.